# Structural evidence for Scc4-dependent localization of cohesin loading

Stephen M Hinshaw[1], Vasso Makrantoni[2], Alastair Kerr[2], Adèle L Marston[2], Stephen C Harrison[1,3]*

[1]Department of Biological Chemistry and Molecular Pharmacology, Harvard Medical School, Boston, United States; [2]Wellcome Trust Centre for Cell Biology, School of Biological Sciences, University of Edinburgh, Edinburgh, United Kingdom; [3]Howard Hughes Medical Institute, Harvard Medical School, Boston, United States

**Abstract** The cohesin ring holds newly replicated sister chromatids together until their separation at anaphase. Initiation of sister chromatid cohesion depends on a separate complex, Scc2$^{NIPBL}$/Scc4$^{Mau2}$ (Scc2/4), which loads cohesin onto DNA and determines its localization across the genome. Proper cohesin loading is essential for cell division, and partial defects cause chromosome missegregation and aberrant transcriptional regulation, leading to severe developmental defects in multicellular organisms. We present here a crystal structure showing the interaction between Scc2 and Scc4. Scc4 is a TPR array that envelops an extended Scc2 peptide. Using budding yeast, we demonstrate that a conserved patch on the surface of Scc4 is required to recruit Scc2/4 to centromeres and to build pericentromeric cohesion. These findings reveal the role of Scc4 in determining the localization of cohesin loading and establish a molecular basis for Scc2/4 recruitment to centromeres.

## Introduction

Tight association between sister chromatids is crucial for successful chromosome segregation in eukaryotic cell division. Cohesin, a ring-shaped protein complex that wraps around sister chromatids (*Gruber et al., 2003*; *Haering et al., 2008*) is the molecular agent of sister chromatid cohesion, which persists from the time of DNA replication until anaphase. A distinct protein complex containing Scc2$^{NIPBL}$ and Scc4$^{Mau2}$ (Scc2/4) initiates linkage of cohesin with DNA (*Ciosk et al., 2000*), and an in vitro reconstitution of cohesin loading by Scc2/4 suggests that the product of this reaction is a topological protein–DNA linkage (*Murayama and Uhlmann, 2013*). In metazoans, cohesin deposition by Scc2/4 is required for normal development (*Dorsett et al., 2005*; *Kawauchi et al., 2009*), and mutations in NIPBL, the human homolog of Scc2, are dominant and causally linked to a severe developmental disorder, Cornelia de Lange Syndrome (CDLS) (*Krantz et al., 2004*).

In addition to initiating a connection between cohesin and chromatin, Scc2/4 determines the timing and location of cohesin loading (*Ciosk et al., 2000*; *Kogut et al., 2009*). Cohesin enrichment at mitotic centromeres and pericentromeres results in tension across sister chromatids when paired kinetochores attach to opposite spindle microtubules (*Tanaka, 2000*). Mitotic spindle checkpoint signaling senses the tension between sister centromeres to ensure correct kinetochore–microtubule attachments (*Stern and Murray, 2001*). Defective centromeric cohesion therefore leads to elevated rates of chromosome missegregation (*Eckert et al., 2007*; *Fernius and Marston, 2009*).

Centromeric cohesion depends on recruitment of Scc2/4 to centromeres in late G1/early S phase (*Hu et al., 2011*; *Fernius et al., 2013*). A group of conserved kinetochore proteins—the Ctf19 complex in yeast (homologous to the human CCAN)—participates in this recruitment pathway, along with the S phase kinase complex, DDK (*Fernius and Marston, 2009*; *Hu et al., 2011*; *Natsume et al., 2013*).

*For correspondence: harrison@crystal.harvard.edu

**eLife digest** DNA replication copies the genetic information contained in a cell's chromosomes. A ring-like protein complex, cohesin, holds together each pair of newly-replicated chromosomes, known as 'sister chromatids'. When the cell divides, cohesin is cleaved; this allows sister chromatids to separate, so that each daughter cell receives one member of each sister chromatid pair and thereby inherits a full complement of genes. Defects in this process result in severe developmental abnormalities. Moreover, the genes that underlie this process are among the most frequently mutated in cancer.

Cohesin is enriched at centromeres: the chromosomal points of attachment to the apparatus (called the 'mitotic spindle') that segregates the sister chromatids into the daughter cells. The protein complex that loads cohesin onto chromosomes determines this preferential localization. The two components of the loading complex, Scc2 and Scc4, associate as a 1:1 pair.

Hinshaw et al. used a technique called X-ray crystallography to determine the structure of Scc4 bound with a large fragment of Scc2. The result showed that the elongated Scc4 twists around the fragment of Scc2, forming an extended groove. Scc2 snakes through the Scc4 groove and emerges at both ends.

Hinshaw et al. then performed a series of experiments in yeast cells to probe how Scc4 determines the location at which cohesin loads onto chromosomes. These experiments revealed that a region on the surface of Scc4 targets both Scc4 and Scc2 to centromeres. The amino-acid sequence of this centromere-targeting patch on the surface of Scc4 is conserved across species. Thus, the mechanism by which Scc4 localizes cohesin to centromeres may be similar in all eukaryotic organisms.

Deletion of any of several Ctf19 complex members leads to impaired centromeric cohesion and chromosome missegregation (*Eckert et al., 2007*; *Fernius and Marston, 2009*; *Ng et al., 2009*; *Hu et al., 2011*), but whether individual components make direct contact with Scc2/4 is not yet known.

The cohesin loading activity of Scc2/4 in vitro requires only Scc2 (*Murayama and Uhlmann, 2013*). Scc4 is essential in yeast, however, and in humans, de novo Mau2$^{Scc4}$ missense and nonsense mutations are significantly underrepresented in exome sequences, indicating that disruption of Mau2$^{Scc4}$ function is probably dominant and lethal (*Table 1*). Thus, Scc2 activity in vivo must depend on Scc4 in ways not recapitulated by the in vitro loading reaction. We report here the structure of yeast Scc4 in complex with an N-terminal fragment of Scc2 and demonstrate that Scc4 determines cohesin localization through a conserved patch on its surface. These findings show that Scc4 targets cohesin loading to a specific genomic locus and that this function is separable from its essential role in establishing sister chromatid cohesion across the genome.

## Results

### Structure of an Scc2$^{1–181}$/Scc4 complex

We prepared full-length Scc2/4 by co-expressing both proteins in baculovirus-infected insect cells. Scc2 is a 1493 amino acid residue protein predicted to have an unstructured N-terminal segment and a C-terminal HEAT (Huntington, EF3, PP2A, TOR1)/ARM (Armadillo) repeat domain (*Figure 1A*). Scc4, also conserved among nearly all eukaryotes, is predicted to have an extensive TPR (TetratricoPeptide Repeat) architecture. Negative stain electron microscopy of the full-length complex showed two large globular structures, variably positioned relative to each other, which we interpret as corresponding to the Scc2 HEAT repeat module and Scc2$^N$-Scc4 (*Figure 1B*). We used limited proteolysis and mass spectrometry to identify an Scc4-containing subcomplex (*Figure 1—figure supplement 1A*). An N-terminal fragment of Scc2 (residues 1–181 or 1–205) is sufficient for stable association with full-length Scc4 (*Figure 1C*). Both the truncated complex of Scc2$^{1–181}$/Scc4 and full-length Scc2/4 are heterodimers in solution (*Figure 1—figure supplement 1B*).

We obtained crystals of full-length Scc4 in complex with an N-terminal, 181-residue fragment of Scc2. Diffraction data collected from selenomethionine-substituted derivatives of these crystals allowed us to determine initial phases by single wavelength anomalous dispersion (SAD). We used an incomplete model, built into a 2.8 Å resolution map, to obtain phase information by molecular

**Table 1**. *De novo* mutation profiles for human NIPBL and Mau2

| | Observed | Expected |
|---|---|---|
| NIPBL | | |
| Synonymous | 58 | 58.4 |
| Missense | 88 | 160.9 |
| Loss of function | 0 | 18.4 |
| Mau2 | | |
| Synonymous | 25 | 26.6 |
| Missense | 13 | 52.7 |
| Loss of function | 0 | 3.8 |

replacement for diffraction data from crystals of a native protein complex, extending to a minimum Bragg spacing of 2.1 Å. Our final model includes residues 5–383, 390–527, and 536–622 of Scc4 and residues 1–64, 73–95, and 106–132 of Scc2 (*Figure 2A*).

Scc4 is a superhelical array of 13 TPR modules with a beta ribbon insertion between repeats 6 and 7 (*Figure 2B*). This tightly wrapped solenoid brings successive turns in contact with each other, and the concave surface becomes an axial groove. Repeat 8, which has a particularly long first helix, lacks a second helix and has instead an extended segment with a disordered surface loop. This irregularity divides the solenoid into two subdomains, $TPR^N$ ($Scc4^{1–384}$) and $TPR^C$ ($Scc4^{391–624}$) (*Figure 2A*).

Residues 10–50 of Scc2 snake along the continuous inner cavity of Scc4 and emerge at both ends, with the N-terminus of Scc2 close to the C-terminus of Scc4. Examples of similar TPR-peptide interfaces include the interaction of the kinesin light chain with cargo peptides (*Figure 2—figure*

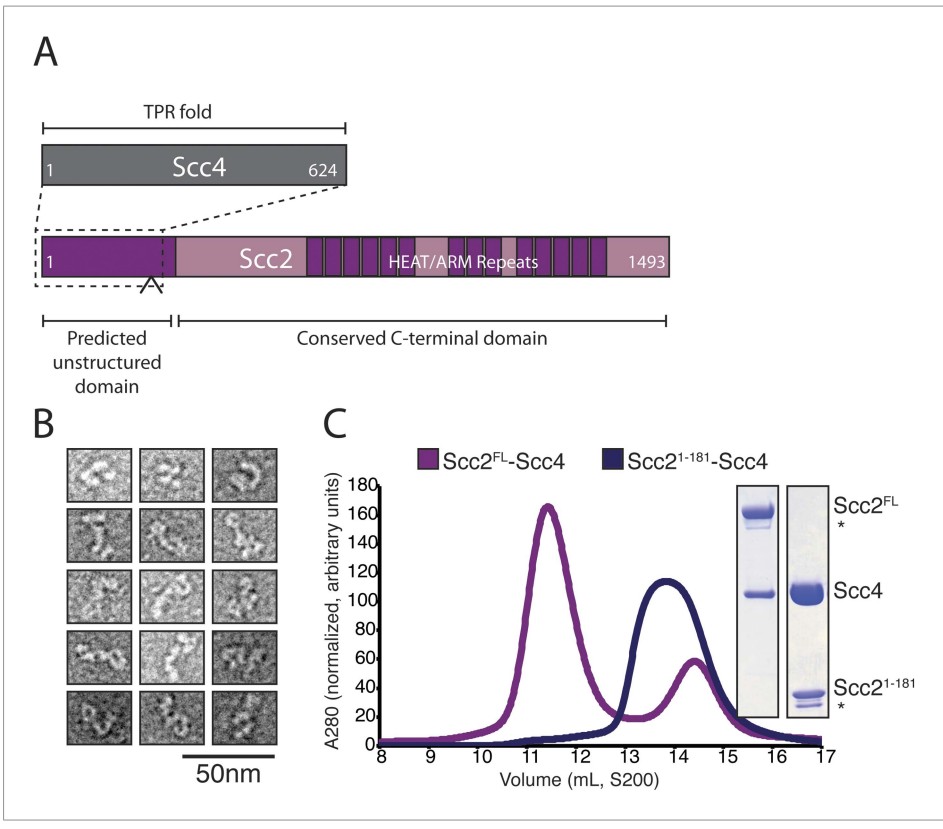

**Figure 1**. Purification of the cohesin-loading complex. (**A**) Domain organization of Scc2/4. Dotted lines show the Scc2–Scc4 interaction. An arrow indicates the position of a regulated cleavage site (*Woodman et al., 2014*). (**B**) Negatively stained Scc2/4 visualized by electron microscopy. Individual particles are shown. (**C**) Gel filtration chromatograms and SDS-PAGE show that $Scc2^{FL}$/Scc4 (magenta, left inset) and $Scc2^{1–181}$/Scc4 (purple, right inset) form stable complexes (* marks an Scc2 cleavage product).

The following figure supplement is available for figure 1:

**Figure supplement 1**. Purification and characterization of an $Scc2^N$–Scc4 complex.

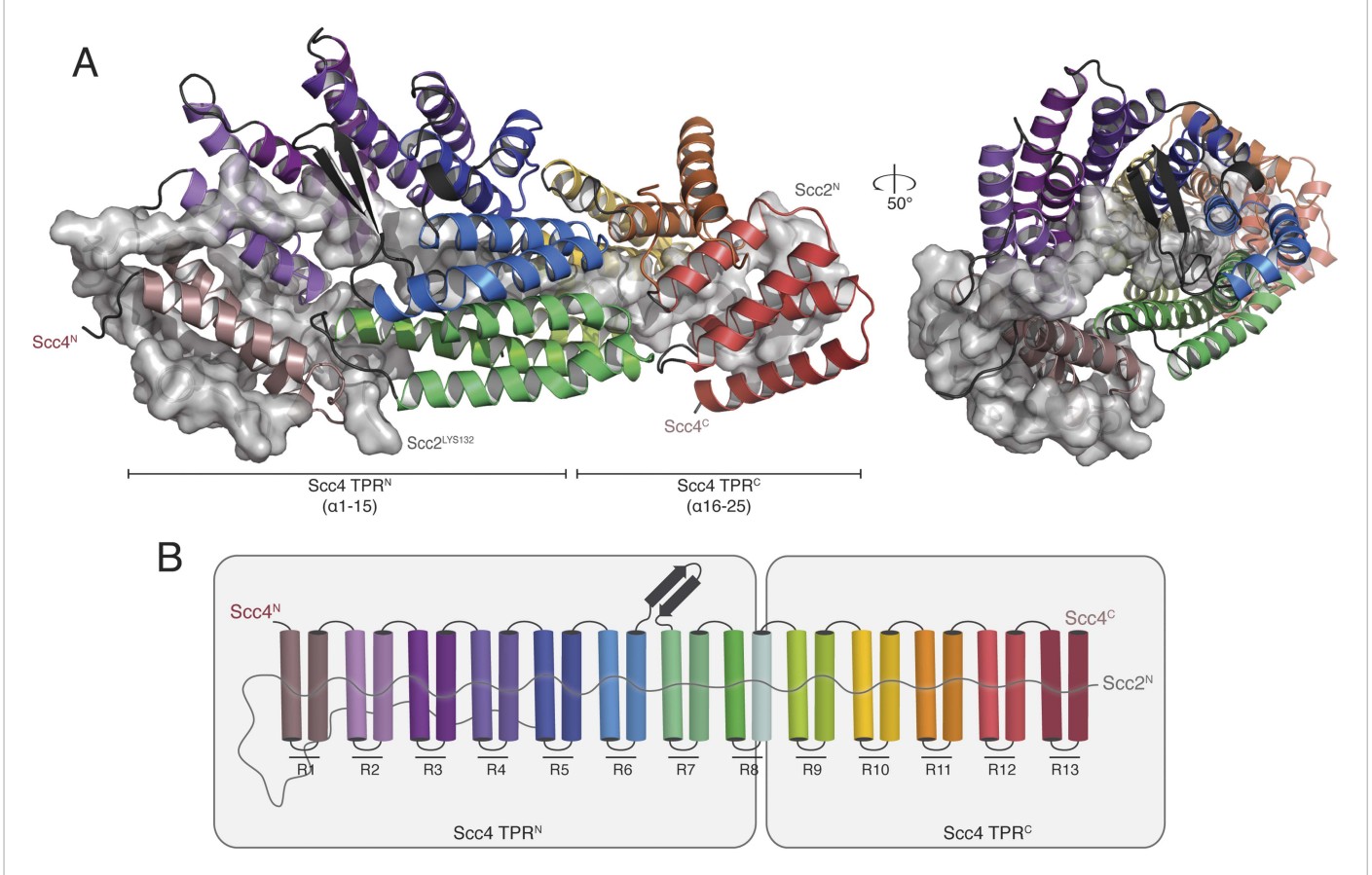

**Figure 2**. Crystal structure of the Scc2$^{1-181}$/Scc4 complex. (**A**) Rotated views of the Scc2$^{1-181}$/Scc4 complex. Scc2 is shown in gray as a cartoon and transparent surface. Individual Scc4 repeats (R1-13) are colored as indicated in (**B**).

The following figure supplements are available for figure 2:

**Figure supplement 1**. Comparison of Scc2$^N$-Scc4 with structural homologs.

**Figure supplement 2**. Structure and conservation of Scc2$^N$.

**Figure supplement 3**. Complementation of Scc4 repression by Scc2.

*supplement 1A*) (*Pernigo et al., 2013*) and the interaction of the cell polarity-determining LGN protein with its binding partners nuclear mitotic apparatus protein 1 (NuMA) and mInscutable (*Figure 2—figure supplement 1B*) (*Zhu et al., 2011*). The extended conformation of Scc2$^{1-181}$ clearly depends on its contacts with the surrounding Scc4; it is likely to be unstructured in the absence of its partner. The Scc2–Scc4 interaction has two unusual features. First, the concave surface of Scc4 is entirely enclosed, and Scc2 dissociation would therefore require either unfolding or proteolysis of Scc2 or Scc4. In fact, regulated proteolysis of Scc2 has been reported recently (*Woodman et al., 2014*). Second, residues 58–132 of Scc2 extend beyond the axial groove and make extensive contact with the external surface of Scc4.

## Conserved Scc2–Scc4 contacts

Human Mau2$^{Scc4}$ binds the N-terminus of human NIPBL$^{Scc2}$ (*Braunholz et al., 2012*), but the primary sequence of NIPBL bears little resemblance to that of Scc2$^{1-181}$. We compiled sequence alignments for the N-terminus of Scc2 from divergent eukaryotes, including yeasts and humans, and mapped amino acid conservation onto its structure (*Figure 2—figure Supplement 2A,B*). Despite their low level of

overall sequence conservation, Scc2/NIPBL proteins have conserved amino acid residues at buried positions that contact Scc4. Several Scc2–Scc4 contacts on the external face of Scc4 are also conserved, including Scc2$^{F86}$, which stacks onto Scc4$^{Y40-Y41}$, and Scc2$^{112–120}$, which forms a helix that fits over a hydrophobic surface on Scc4 TPR$^N$. We found that Scc2 variants mutated at conserved residues contacting either the concave or the external surface of Scc4 were less effective than wild-type Scc2 in restoring viability to an Scc2-degron (*SCC2-AID*) strain under depletion conditions (*Figure 2—figure supplement 2C*). The phenotype of these mutants was comparable to the phenotype we observed when we removed the entire Scc2 fragment visible in the crystal structure (Δ1–138).

If cohesin loading can occur in the absence of Scc4 in vitro (*Murayama and Uhlmann, 2013*) and to some extent in the absence of the Scc4-binding region of Scc2 in vivo, why is *SCC4* an essential gene? We found that supplementing the chromosomal copy of *SCC2* with a plasmid coding for Scc2 or Scc2$^{Δ1–138}$ complemented transcriptional repression of *SCC4* (*pGAL1-SCC4*). That is, increasing the *SCC2* gene copy number bypassed the effect on viability of transcriptional repression of *SCC4*, and the bypass did not require the Scc4-binding region of Scc2 (*Figure 2—figure supplement 3*). Because this region of Scc2 is probably unstructured when not bound by Scc4, it is a plausible cause of instability or aggregation when expressed unprotected. One reason why Scc4 is essential may therefore be that Scc2 is unstable in its absence.

## A conserved surface patch on Scc4

Mapping Scc4 sequence conservation onto our structure revealed a cluster of extremely conserved, solvent-facing residues, some of them invariant among all eukaryotes we examined (*Figure 3A,B*). We found unaccounted-for electron density in this patch, into which we modeled a sulfate group. Because bound sulfates often mark phosphate binding sites in crystal structures (*Bax et al., 2001*), we tested the effects of mutating conserved residues that contribute to this patch. Strains in which the endogenous copy of *SCC4* had been replaced by *SCC4* coding for mutations at these residues were viable but displayed a plasmid missegregation phenotype (*Figure 3—figure supplement 1A*). Combined mutation of seven of the most conserved residues (*scc4$^{L256L; Y298A; K299D; Y313A; F324A; K327D; K331D}$*; *scc4$^{m7}$*) did not inactivate Scc4, as strains bearing these substitutions were fully viable and had a plasmid segregation phenotype comparable in strength to that of strains bearing mutations only at positions 324, 327, and 331 (*scc4$^{F324A; K327D; K331D}$*; *scc4$^{m3}$*) (*Figure 3—figure supplement 1B,C*). Recombinant Scc2$^{1–181}$-Scc4 complexes bearing these mutations behaved identically to wild-type preparations (*Figure 1—figure supplement 1B*), indicating that perturbation of the conserved Scc4 patch does not impair the stability or folding of the rest of the protein.

Strains lacking the Ctf19 complex subunit Chl4 (CENP-N in humans) are defective in centromeric cohesin loading because they cannot preferentially localize Scc2/4 to centromeres, although cohesin loads normally elsewhere in the genome (*Fernius et al., 2013*). To determine whether the Scc4-conserved patch functions in the same pathway as Chl4, we introduced the *scc4$^{m3}$* mutation into a *chl4Δ* strain and tested for its effect on plasmid segregation (*Figure 3C*). Inclusion of *scc4$^{m3}$* does not augment the severe plasmid loss phenotype caused by *CHL4* deletion. This relationship also holds true for *scc4$^{m7}$* (*Figure 3—figure supplement 1B*). Strains lacking Chl4 have extended metaphase spindles, and this phenotype corresponds to weakened centromeric cohesion (*Fernius and Marston, 2009*; *Laha et al., 2011*). We found that an *scc4$^{m3}$*-bearing strain exhibited increased inter-spindle pole distances and that the *scc4$^{m3}$* mutation did not exacerbate the spindle extension phenotype of a *chl4Δ* strain (*Figure 3D*). Moreover, GFP-labeled sister centromeres (*Figure 4—figure supplement 1A*), but not chromosome arms (*Figure 4—figure supplement 1D–G*), were separated more frequently and to greater distances in strains mutated at 5 positions in the conserved patch on Scc4 (*scc4$^{F324A; K327A; K331A; K541A; K542A}$*; *scc4$^{m35}$*). We conclude that the conserved Scc4 patch promotes centromeric cohesion and that it does so in a manner that may also depend on *CHL4*.

## Scc4 mutations disrupt centromeric cohesin loading

These results suggest that interactions at the Scc4-conserved patch target Scc2/4 specifically to centromeres. To test this hypothesis, we measured Scc2 and cohesin localization by chromatin immunoprecipitation (ChIP). Perturbation of the conserved patch (either *scc4$^{m35}$* or *scc4$^{m7}$*) eliminates centromeric localization of Scc2 in mitotic cells and reduces association of the cohesin subunit, Scc1, with the centromere and pericentromere, but not with chromosome arms (*Figure 4B,C*, *Figure 4—figure supplement 2A–D*). We also observed this pattern of Scc2 localization in cells

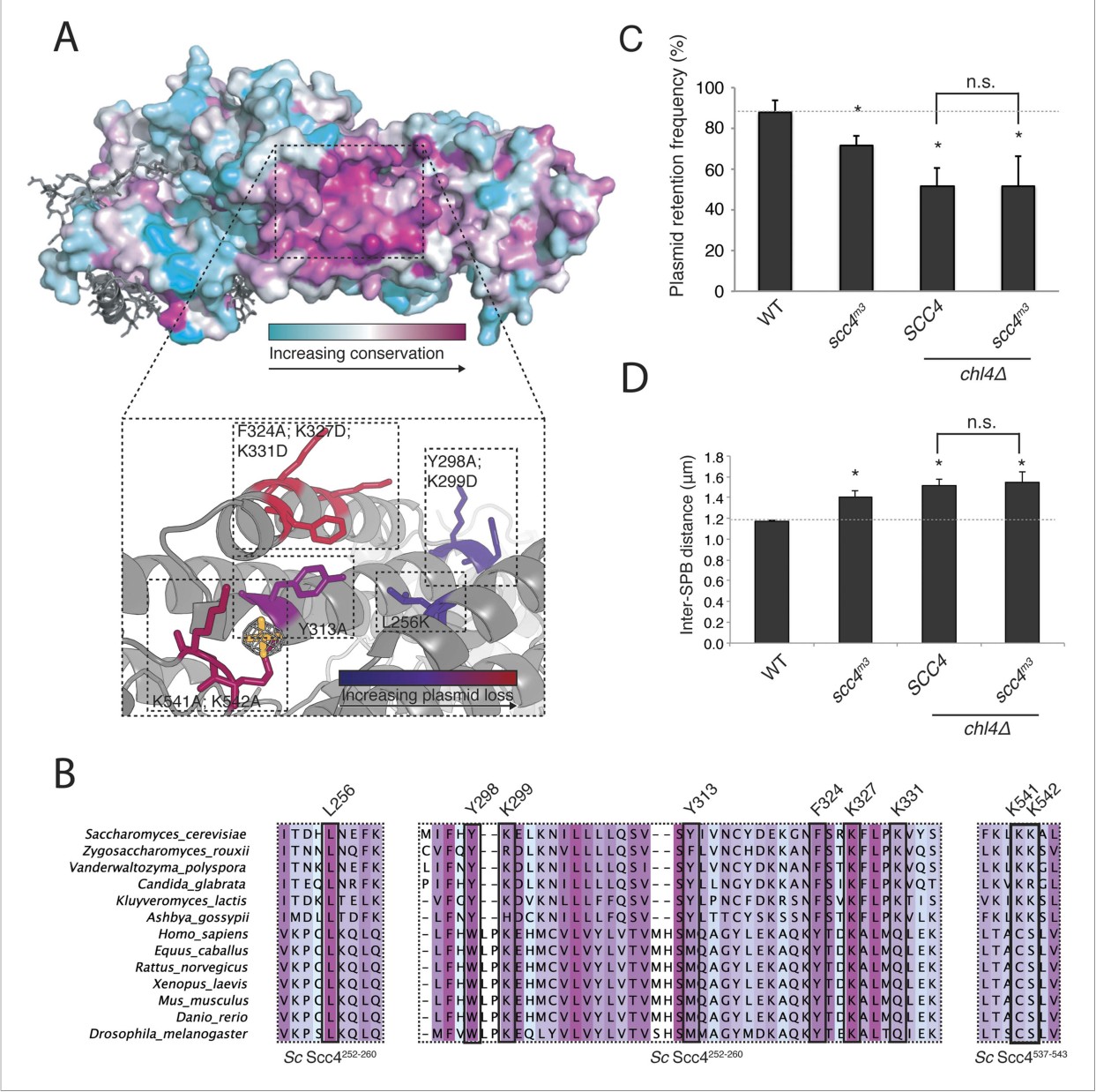

**Figure 3**. A conserved patch on the surface of Scc4. (**A**) Surface view of Scc4 colored according to primary sequence conservation across eukaryotes. Inset shows the Scc4-conserved patch with mutated residues labeled and colored according to their effect on plasmid segregation fidelity (*Figure 3—figure supplement 1*). (**B**) Multiple sequence alignment of Scc4 and homologs from fungi and metazoans. Alignment is colored by conservation according to the color scheme in (**A**). (**C**) Plasmid missegregation was measured for the indicated strains ($scc4^{m3}$—$scc4^{F324A; K327D; K331D}$; error bars indicate SD; * $p < 0.05$, Student's t-test vs WT, two tails; n.s. indicates $p > 0.05$.). The dotted line shows the rate of plasmid segregation in a WT background. (**D**) Spindle length measurements for each indicated strain arrested in S phase with hydroxyurea. The dotted line shows the WT mean spindle length (error bars indicate SD; * $p < 0.05$, Student's t-test vs WT, two tails).

The following figure supplement is available for figure 3:

**Figure supplement 1**. (**A**) Plasmid segregation defects in Scc4-conserved patch mutants.

progressing through S phase (*Figure 4—figure supplement 2E,F*), the stage at which cohesin loading is initiated (*Kogut et al., 2009*; *Natsume et al., 2013*).

We further analyzed Scc1 localization by ChIP followed by high-throughput sequencing (ChIP-seq) and found that all 16 centromeres were specifically depleted of Scc1 in the $scc4^{m35}$ background

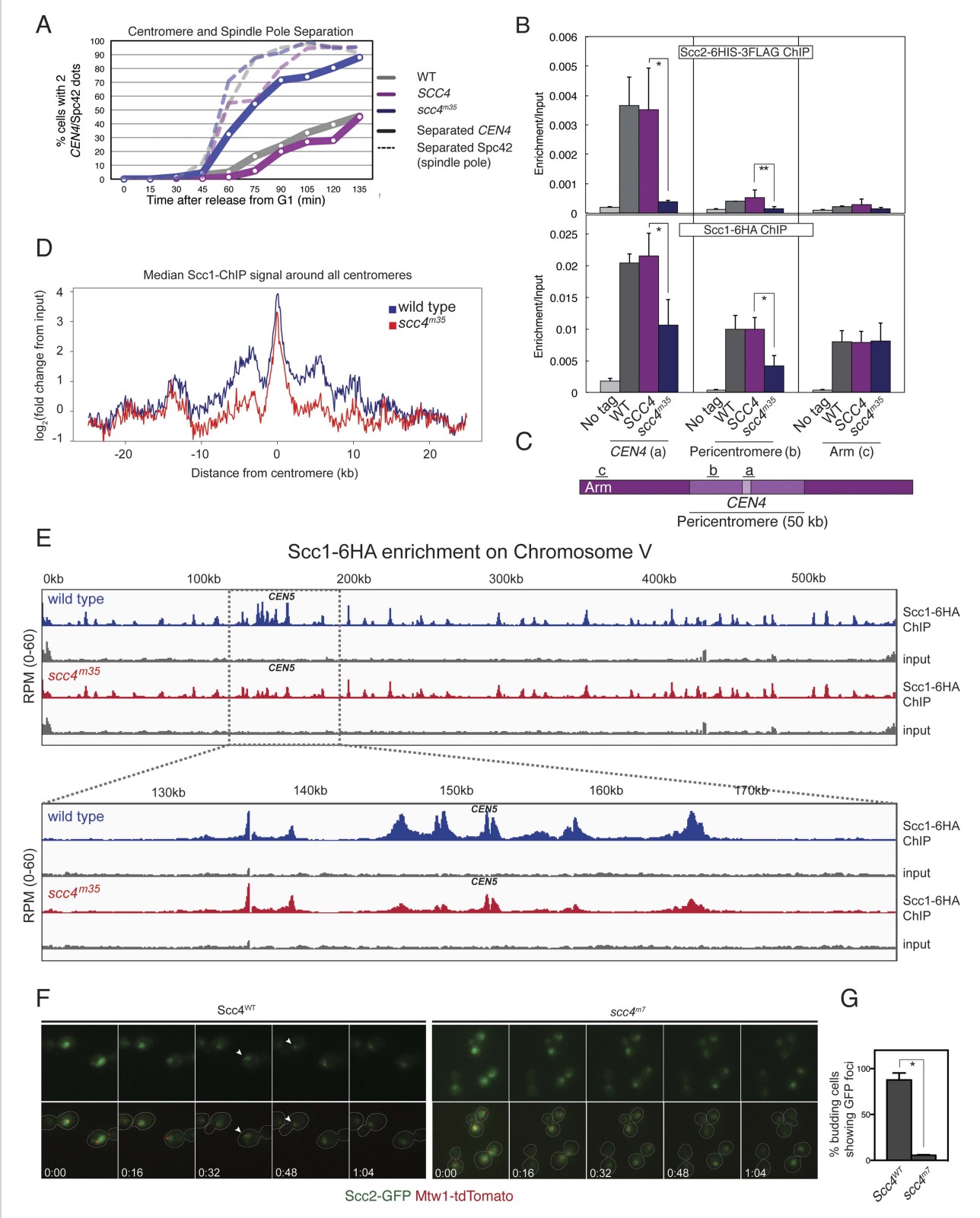

**Figure 4.** Defective centromeric cohesin loading in Scc4 conserved patch mutants. (**A**) Sister centromeres (+2.4CEN4-GFP) are separated earlier and more frequently in an scc4[m35] strain entering the cell cycle after a G1 arrest. pMET-CDC20 strains of the indicated genotypes (gray—wild type; magenta—SCC4 integrated; blue—scc4[m35] integrated) were arrested in G1 with alpha factor and released into the cell cycle in the presence of methionine to repress

*Figure 4. continued on next page*

*Figure 4. Continued*

*CDC20* expression. Solid lines show the percent of cells with separated *CEN4* dots, and dotted lines show the percent of cells with separated spindle pole bodies (Spc42-tdTomato). (**B**) Strains of the indicated genotypes and either Scc2-His6-3FLAG (top) or Scc1-6HA (bottom) were arrested in metaphase of mitosis following treatment with nocodazole and benomyl (to depolymerize microtubules). Cells were harvested after 2 hr. Anti-FLAG or anti-HA antibodies were used for ChIP, and pulldown samples were analyzed by qPCR. Mean values of four independent experiments are shown (error bars indicate ±SD; * p < 0.05; **p < 0.01 paired two-tailed t-test). (**C**) Schematic of a fragment of chromosome IV showing the location of qPCR amplicons used in (**B**). (**D**) Scc1 enrichment in a 50-kb domain surrounding all 16 budding yeast centromeres is shown for wild-type and $scc4^{m35}$ cells. For both wild type and $scc4^{m35}$, the ratio of reads (normalized to RPM) over input in a 100-bp window was calculated separately for each chromosome at the indicated position. The median count value for each window was then plotted to give a composite view of all 16 pericentromeres. (**E**) Scc1 enrichment along chromosome V together with a magnification of a 50-kb region including the centromere is shown. The number of reads at each position was normalized to the total number of reads for each sample (RPM: reads per million) and shown in the Integrated Genome Viewer from the Broad Institute (***Robinson et al., 2011***). (**F**) Live cell imaging of homozygous diploid cells expressing Scc2-GFP, Mtw1-tdTomato to mark centromeres, and the indicated version of Scc4 (left—wild type; right $scc4^{m7}$). GFP dots observed in wild-type cells are marked with white arrows in the first frame in which they are visible, and these foci were not observed in $scc4^{m7}$ cells. Time is given relative to the start of the imaging session (hr:mm). (**G**) Quantification of live cell imaging. At least 10 budding cells per field (three fields of view each for each strain for three separate experiments) were scored for the presence of an Scc2-GFP focus (error bars indicate ±SD; *p < 0.005, two-tailed t-test).

The following figure supplements are available for figure 4:

**Figure supplement 1**. Spindle pole and CEN separation but not chromosome arm separation in an Scc4-conserved patch mutant.

**Figure supplement 2**. Scc2 and Scc1 association with centromeres and chromosome arms.

**Figure supplement 3**. Scc1 is reduced around all 16 individual centromeres in $scc4^{m35}$ cells.

(*Figure 4—figure supplement 3*). Scc1 depletion extended roughly 10 kilobases to either side of the core centromere but not to chromosome arms (*Figure 4D,E*). Moreover, centromeric Scc2-GFP signal, normally visible in wild-type cells and eliminated by *CHL4* deletion (*Fernius et al., 2013*), was lost upon mutation of the Scc4-conserved patch (*Figure 4F,G*). These results indicate that, like Chl4, a specific interaction surface on Scc4 is required to target cohesin loading to centromeres.

## Discussion

Cohesin targeting to specific genomic locations is required in all species for robust centromeric cohesion and for tissue-specific transcriptional programs in multicellular organisms (*Kawauchi et al., 2009*; *Fay et al., 2011*). DNA sequences per se do not drive cohesin loading (*Onn and Koshland, 2011*; *Murayama and Uhlmann, 2013*). Instead, targeting likely depends on interactions between the loading complex and chromatin-associated factors. We have determined the structure of Scc4 bound to a minimal fragment of Scc2, and we have used this structure to derive separation of function alleles that uncouple cohesin loading from cohesin targeting to centromeres. These experiments demonstrate that cohesin targeting in yeast depends on Scc4. This finding is consistent with unpublished genetic evidence for an Scc4-dependent cohesin localization pathway (Nasmyth, personal communication). Because the pathway that targets cohesin to centromeres requires Scc4 residues that are, in some cases, invariant across diverse eukaryotes, we suggest that the Scc4-dependent cohesin targeting we describe is a general feature of the control of cohesin loading.

Consistent with previous reports (*Fernius et al., 2013*; *Natsume et al., 2013*), we find that the centromeric enrichment of cohesin loading is not essential for viability and that arm cohesion is unaffected in strains in which this pathway is compromised. These results probably reflect two modes for cohesin loading: one that depends on a conserved patch of Scc4 and happens at centromeres, and a second opportunistic mode that happens everywhere on the chromosome and is sufficient to support viability in the absence of the first mode. If so, Scc4-conserved patch mutations would not eliminate cohesin loading but would redistribute loading events, resulting in similar cohesin levels at centromeres and on chromosome arms. This prediction is borne out both by strains bearing Scc4-conserved patch mutations and by strains lacking *CHL4* (*Fernius and Marston, 2009*).

Recent reports have shown that cohesin targeting to specific locations along the chromosome arms of fission yeast (*Mizuguchi et al., 2014*) and flies (*Oliveira et al., 2014*) is critical for normal three-dimensional chromosome structure. In addition to findings from studies conducted in mammalian cells

(*Dowen et al., 2014*), one of these reports (*Mizuguchi et al., 2014*) suggests that cohesin complexes residing at specified sites in the genome function in gene looping and in defining chromosome territories. Because our ChIP-seq experiments show that Scc4 directs cohesin localization to broad centromeric domains but does not affect the local distribution of cohesin peaks, we suggest that the Scc4-dependent localization pathway we discuss here is overlaid upon local determinants of cohesin positioning on chromosomes. These local determinants would be a second level of cohesin regulation, operating at roughly the resolution of the transcriptional units in yeast. Integration of these spatial cues with temporal cues, including the kinase activity of DDK, could then generate the final cohesin distribution observed in metaphase cells. The Scc4-conserved patch presents a possible explanation for how broad cohesin-dense domains may be specified by the cohesion-loading complex, and it provides a molecular foundation for the study of chromatin-associated factors, both known and unknown, that localize cohesin loading.

## Materials and methods

### Exome analysis
Analysis of exome data sets was performed as described (*Samocha et al., 2014*).

### Plasmids
Coding sequences for Scc2 and Scc4 were amplified from yeast genomic DNA and inserted into a modified version of pFastbac (Life Technologies, Carlsbad, CA) suitable for ligation-independent cloning. The expression vector contains an N-terminal 6-His tag followed by a TEV protease sequence. Full-length Scc4 and fragments of the Scc2 coding sequence were cloned by the same procedure into a bacterial expression vector for expression from a single mRNA. The coding sequences for both genes were augmented such that each contains an N-terminal 6-His tag followed by a TEV protease cleavage site.

For complementation experiments, the Scc2 or Scc4 locus was amplified from *Saccharomyces cerevisiae* genomic DNA by PCR and cloned by restriction digest into a plasmid containing a CEN-ARS cassette and a selectable auxotrophic marker (LEU2). Genomic regions included were as follows: Scc2—chrIV:820797–825978; Scc4—chrV:465380–462755. We used PCR stitching and isothermal assembly to generate mutated versions of these constructs.

### Yeast strains and culture conditions
Yeast strains bearing Scc4 point mutations or gene deletions were constructed using PCR methods as previously described (*Longtine et al., 1998*). All Scc4-conserved patch mutations described in this text were achieved by replacement of the native *SCC4* locus. Viability of s288c strains expressing *SCC4* mutants was first confirmed by complementation of *pGAL1-SCC4* repression as shown for *Figure 2—figure supplement 3* using plasmids bearing the *SCC4* chromosomal locus. Viability of the *scc4^{m7}* strain was determined by FACS analysis of homozygous diploid cells and during strain construction by sequential sporulation of the diploid imaging strains. To generate w303 strains carrying *scc4* alleles, diploid strain (AM14499) carrying a heterozygous deletion (*scc4Δ::KanMX6*) was transformed with a PCR product corresponding to full-length *SCC4* (or its mutant derivatives) and a downstream marker (*HIS3*). G418-sensitive, histidine prototrophs were sporulated, and *SCC4* mutations were confirmed in the haploids by sequencing. All Scc2-GFP strains are derivatives of Scc2-GFP from the Yeast GFP Clone Collection (*Huh et al., 2003*). Strains for live cell imaging were constructed by integration of an Mtw1-tdTomato PCR into the Scc2-GFP strain followed by mating to achieve the final diploid strains. Plasmid segregation experiments were performed essentially as described previously (*Hinshaw and Harrison, 2013*).

Auxin-inducible degron-tagged Scc2 (*SCC2-AID*) was generated using PCR methods (*Nishimura et al., 2009*). For complementation assays, *SCC2-AID* cells bearing a CEN-ARS (Chr VI) plasmid encoding Scc2 flanked by its native control elements were grown to mid-log phase in synthetic complete (SC) medium lacking leucine (to select for the plasmid). Cells were plated in a fivefold dilution series on a solid SC medium lacking leucine and supplemented with the indicated amount of 1-Naphthaleneacetic acid (Auxin; Sigma-Aldrich, St. Louis, MO). Benomyl and nocodazole were used at 30 μg/ml and 15 μg/ml, respectively.

## Protein expression and purification

Recombinant baculoviruses for His6-Scc2 and His6-Scc4 were amplified separately in Sf21 cells (Life Technologies) for three passages. For protein expression, *Trichoplusia ni* cells grown in suspension in Ex-Cell405 medium (Sigma–Aldrich) were infected with equal amounts of both viruses, and cells were pelleted and resuspended for freezing in lysis buffer (40 mM HEPES pH 7.5, 20 mM imidazole, 50 mM NaCl, 10% glycerol, and 2 mM β-mercaptoethanol) after 72 hr. Upon thawing and addition of protease inhibitors (4 μM aprotinin, 1 μM leupeptin, 1.4 μM pepstatin, and 1 mM PMSF), NaCl was added to a final concentration of 800 mM, and cells were broken by Dounce homogenization and sonication. Insoluble material was pelleted by centrifugation for 30 min at 18,000 rpm in a JA-20 rotor (Beckman-Coulter, Pasadena, CA). 6-His-tagged Scc2/4 complexes were isolated from the supernatant by $Co^{2+}$ affinity chromatography followed by ion exchange chromatography (HiTrap SP HP, GE Healthcare, UK) and size exclusion chromatography (Superdex 200 20/16, GE) in gel filtration buffer (20 mM Tris–HCl pH 8.5, 200 mM NaCl, 1 mM TCEP).

To isolate $Scc2^N$–Scc4 complexes, polycistronic expression vectors (described above) encoding these proteins under the control of a single T7 promoter were transformed into the *Escherichia coli* strain Rosetta 2(DE3)pLysS (Millipore, Billerica, MA), and protein expression was induced with 400 μM IPTG at an OD600 of approximately 0.5. Bacterial cultures were further incubated overnight at 18°C. Cells were resuspended and frozen in lysis buffer containing 800 mM NaCl. Upon thawing and addition of protease inhibitors, cells were lysed by sonication, and 6-His-tagged proteins were purified as described for full-length Scc2/4. Selenomethionine-derivatized (SeMet) $Scc2^{181}$-Scc4 samples were prepared as described for native protein samples with the exception that growth medium was prepared as described previously (*Hinshaw and Harrison, 2013*).

## Electron microscopy and SEC-MALS

Purified Scc2/4 was diluted in gel filtration buffer and adsorbed to glow discharged carbon-coated copper grids. After staining with 0.75% (wt/vol) uranyl formate, grids were imaged using a CM10 electron microscope (Philips, Amsterdam).

For size exclusion chromatography coupled to multiple angle light scattering (SEC-MALS), experiments were performed essentially as described previously (*Hinshaw and Harrison, 2013*) with the exception that a 3-ml size exclusion column was used for analysis of truncated $Scc2^N$–Scc4 complexes (Superdex 200 5/150 GL; GE Healthcare).

## Crystallization and structure determination

Crystals of $Scc2^{1–181}$-Scc4 formed overnight at 18°C. For native crystals, the protein was concentrated in gel filtration buffer to 18 mg/ml and mixed in a 1-to-1 ratio (vol:vol) with crystallization buffer (0.2M ammonium sulfate, 16% [wt:vol] PEG 3350). Crystals were washed first in wash buffer (160 mM NaCl, 16 mM Tris–HCl pH 8.5, 1 mM TCEP, 14.4% PEG 3350, and 0.16M ammonium sulfate) and then in wash buffer supplemented with 30% (vol:vol) glycerol before flash freezing in liquid nitrogen. SeMet-derivative crystals were concentrated to 18 mg/ml, and diffracting crystals formed in crystallization buffer with 20% (wt:vol) PEG 3350. These crystals were frozen as described for native versions. All diffraction data were collected on NE-CAT beamline 24ID-E. Data were indexed and scaled with XDS (SeMet) (*Kabsch, 2010*) or HKL2000 (native data) (*Otwinowski and Minor, 1997*).

The structure of $Scc2^{1–181}$-Scc4 was initially determined by SAD. To locate selenium atoms, we used SHELXD as implemented by HKL2MAP (*Pape and Schneider, 2004*). A search for 20 heavy atom sites with a resolution cutoff of 4 Å yielded a solution with 28 heavy atom positions. A truncated list of coordinates for 18 heavy atoms was used to generate an initial map at 3 Å resolution using Phenix Autosol (*Adams et al., 2010*). After density modification using Resolve (as implemented by Phenix), the map displayed extensive density corresponding to alpha helices. Placement of ideal helices and refinement using Phenix Refine yielded a partial structure, which was then used as a search model for phase determination by molecular replacement using a high-resolution native data set and Phaser-MR. During later stages of refinement, riding hydrogens were included, and TLS groups were invoked for Scc4 (*Painter and Merritt, 2006*). Crystallography statistics are shown in *Table 2*, and the coordinates have been deposited in the Protein Data Bank, accession number 4XDN.

**Table 2**. Crystallographic data collection and refinement statistics

| | Scc2$^{1-181}$; Scc4 (SeMet) | Scc2$^{1-181}$; Scc4 (Native) |
|---|---|---|
| **Data collection** | | |
| Resolution (Å) | 30.0–2.8 | 178–2.0 |
| Wavelength (Å) | 0.979210 | 0.979240 |
| Space group | P2$_1$2$_1$2$_1$ | P2$_1$ |
| Unit cell dimensions (a, b, c) (Å) | 58.6, 89.0, 178.0 | 51.9, 178.1, 52.7 |
| Unit cell angles (α, β, γ) (°) | 90, 90, 90 | 90, 111.7, 90 |
| I/σ (last shell) | 11.6 (1.9) | 6.0 (1.3) |
| R$_{sym}$ (last shell) (%) | 14.1 (92.3) | 11.0 (72.9) |
| Completeness (last shell) (%) | 99.7 (90.0) | 93.0 (89.7) |
| Number of reflections | 168241 | 154940 |
| unique | 23460 | 50878 |
| Number of Se sites | 18 | – |
| **Refinement** | | |
| Resolution (Å) | – | 28.7–2.1 |
| Number of reflections | – | 47188 |
| working | – | 45322 |
| free | – | 1866 |
| R$_{work}$ (last shell) (%) | – | 18.5 (28.7) |
| R$_{free}$ (last shell) (%) | – | 21.0 (28.0) |
| **Structure Statistics** | | |
| Number of atoms (protein) | – | 5845 |
| sulfate | – | 24 |
| solvent | – | 301 |
| r.m.s.d. bond lengths | – | 0.004 |
| r.m.s.d. bond angles | – | 0.661 |

## Spindle pole separation assay

Log-phase cultures grown in SC medium were arrested in S phase with 10 mg/ml hydroxyurea for 90 min. Cells were fixed at room temperature for 10 min with 3.7% formaldehyde, washed twice with phosphate buffered saline, pH 8.5, and resuspended in wash buffer containing 1.2 M sorbitol. Cells were immobilized on concanavalin-A-coated cover slips and imaged using a Nikon Ti motorized inverted microscope with a 60× objective lens (NA 1.4) and a Hamamatsu ORCA-R2 cooled digital camera. Z-stacks (11 × 0.3 μm) were acquired with MetaMorph image acquisition software, and maximum z-projections were generated with ImageJ.

To calculate spindle pole distances, we wrote a Matlab script that identifies Spc110-mCherry foci and calculates a distance to the nearest neighbor for each instance. The list of distances was filtered to remove redundant measurements and to remove measurements arising from S phase spindles that straddled the edge of the image (distance measurements surpassing 32.9 μm).

## CENIV dot and Spc42 separation

Cell growth and measurements were carried out as described previously (*Fernius et al., 2013*). Strain genotypes are listed in the strain table (*Supplementary file 1*).

## Live cell imaging

Cells were grown in an SC medium overnight and diluted 1:20 (vol:vol) the next morning. After 6 hr, cells were immobilized on concanavilin A-coated cover slips and a fresh SC medium was applied.

Experimental and control strains were loaded in adjacent imaging chambers, and cells were maintained at 30℃ with high humidity using a Tokai Hit stage top incubator. Live cell images were captured using the imaging setup described above with the exception that Z-stacks ($8 \times 0.3$ μm) were acquired every 8 min. We used exposure times of 10 ms (tdTomato) and 200 ms (GFP) for each image. Maximum intensity projections were generated with ImageJ for each timepoint, and figures were created with Nikon Elements software using identical processing steps and settings for each image.

## Chromatin immunoprecipitation and qPCR

ChIP-qPCR and sequencing experiments were carried out as described previously (*Fernius et al., 2013*; *Verzijlbergen et al., 2014*). Scripts, data files, and workflows used to create the ChIP-Seq figures can be found on the github repository at https://github.com/AlastairKerr/Hinshaw2015. ChIP-Seq data sets have been deposited with the NCBI Gene Expression Omnibus under the accession number GSE68573.

## FACS analysis

Flow cytometry was performed as previously described (*Fernius et al., 2013*). 5,000 cells were analyzed for each sample.

## Acknowledgements

We thank Jonathan Schuermann and the staff at NE-CAT for help with data collection. We thank Simon Jenni for help with data processing and Kevin Corbett for critical reading of the manuscript. Microscopy experiments were performed with assistance from the Nikon Imaging Center at Harvard Medical School. We thank Bianka Baying at Genecore EMBL for library preparation and sequencing. This work was supported by funding from the National Science Foundation (SMH), HHMI (SCH), and the Wellcome Trust [090903, 092076, 096994].

## Additional information

### Competing interests

SCH: Reviewing editor, *eLife*. The other authors declare that no competing interests exist.

### Funding

| Funder | Grant reference | Author |
|---|---|---|
| Wellcome Trust | 090903 | Vasso Makrantoni, Alastair Kerr, Adèle L Marston |
| National Science Foundation (NSF) | Graduate student fellowship | Stephen M Hinshaw |
| Howard Hughes Medical Institute (HHMI) | | Stephen C Harrison |
| Wellcome Trust | 096994 | Vasso Makrantoni, Alastair Kerr, Adèle L Marston |
| Wellcome Trust | 092076 | Vasso Makrantoni, Alastair Kerr, Adèle L Marston |

The funders had no role in study design, data collection and interpretation, or the decision to submit the work for publication.

### Author contributions

SMH, VM, Conception and design, Acquisition of data, Analysis and interpretation of data, Drafting or revising the article; AK, Analysis and interpretation of data; ALM, SCH, Conception and design, Analysis and interpretation of data, Drafting or revising the article

### Author ORCIDs

Alastair Kerr, http://orcid.org/0000-0001-9207-6050

## Additional files

### Supplementary files

• Supplementary file 1. Yeast strains used in this study.

• Supplementary file 2. Primers for ChIP-qPCR experiments in this study.

### Major datasets

The following dataset was generated:

| Author(s) | Year | Dataset title | Dataset ID and/or URL | Database, license, and accessibility information |
|---|---|---|---|---|
| Kim H, Grunkemeyer TJ, Modi C, Chen L, Fromme R, Matz MV, Wachter RM | 2013 | Crystal Structure of a reconstructed Kaede-type Red Fluorescent Protein, Least Evolved Ancestor (LEA) | http://www.rcsb.org/pdb/explore/explore.do?structureId=4DXN | Publicly available at RCSB Protein Data Bank (Accession No: 4DXN). |

The following previously published datasets were used:

| Author(s) | Year | Dataset title | Dataset ID and/or URL | Database, license, and accessibility information |
|---|---|---|---|---|
| Samocha | 2013 | database of Genotypes and Phenotypes (dbGaP) | http://www-ncbi-nlm-nih-gov.ezp-prod1.hul.harvard.edu/projects/gap/cgi-bin/study.cgi?study_id=phs000298.v1.p1 | Publicly available to query here: http://atgu.mgh.harvard.edu/webtools/gene-lookup/. |
| Pernigo S, Lamprecht A, Steiner RA, Dodding MP | 2013 | Crystal structure of the TPR domain of kinesin light chain 2 in complex with a tryptophan-acidic cargo peptide | http://www.rcsb.org/pdb/explore/explore.do?structureId=3ZFW | Publicly available at RCSB Protein Data Bank (Accession No: 3ZFW). |
| Zhu J, Wen W, Zheng Z, Shang Y, Wei Z, Xiao Z, Pan Z, Du Q, Wang W, Zhang M | 2011 | Structures of the LGN/NuMA complex | http://www.rcsb.org/pdb/explore/explore.do?structureId=3RO2 | Publicly available at RCSB Protein Data Bank (Accession No: 3RO2). |

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
