## [Decision Letter]

Thank you for sending your work entitled “Structural evidence for Scc4-dependent localization of cohesin loading” for consideration at *eLife*. Your article has been favorably evaluated by James Manley (Senior editor) and three reviewers, including Kim Nasmyth and Leemor Joshua-Tor, who is a member of our Board of Reviewing Editors. However, a number of points require attention, as described below.

The Reviewing editor and the other reviewers discussed their comments before we reached this decision, and the Reviewing editor has assembled the following comments to help you prepare a revised submission.

This paper reports on a crystal structure of a complex, between Scc4 and the N-terminus of Scc2. The Scc2/4 complex is essential for loading cohesin onto chromosomes. It has also been implicated in regulating chromosome structure and in controlling gene expression. Haplo-insufficiency of its Scc2 subunit (Nipbl) causes Cornelia de Lange syndrome in humans, an effect thought but not yet proven to be due to subtle defects in cohesin's genomic distribution. It has been proposed but not yet proven that cohesin loading corresponds to entrapment of DNAs by cohesin rings. If so, the function of Scc2/4 must be to facilitate DNA entry, presumably by opening transiently one of the interfaces connecting its Smc1, Smc3, and kleisin subunits. It has been reported that Scc2's ortholog in *S.pombe*, Mis4, can promote the stable association of cohesin with DNA in vitro in the absence of Scc4 but whether this association is truly topological is not known. How then does Scc2/4 facilitate cohesin loading and what is the function of Scc4 if it is not directly involved in the loading reaction? These are questions of central importance for the cohesin field and, because cohesin is just one of several related Smc/kleisin complexes, a topic of broad general interest for the field of chromosome biology.

The reviewers felt that the structure is an important advance to the field; however, the functional experiments are not performed using state of the art methodologies, as will be elaborated upon below, and the conclusions based on them do not appear to be justified by the data presented in the paper. Many of these should be straightforward to rectify and the authors are encouraged to consider resubmission upon completion of the following:

1) More extensive mutagenesis of the surface on Scc4.

2) Since ChIP with Scc2 is unreliable, results should be backed up by live cell imaging, which is more reliable and a better reflection of what is going on in the cell. This should be done using state of the art imaging (see point 8).

3) Whole genome comparisons of ChIP data would be a more rigorous way of demonstrating that Scc4 is concerned with loading cohesin around centromeres and not along arms. Such a conclusion is hard to make merely on the basis of analyzing a few sequences.

Major comments:

The structure reveals that Scc4 forms a superhelical array of TPR modules, as predicted from sequence analyses, and that the N-terminal residues of Scc2 snake along its inner cavity, which is entirely novel information. As might be expected, mutation of conserved residues within Scc2's NTD making contact with Scc4 compromise viability, as does deletion of the entire NTD. This is not unexpected as Scc4 itself is an essential gene. Mapping sequence variation onto the Scc4 structure revealed a cluster of highly conserved surface residues, mainly basic and hydrophobic. One might think therefore that this would prove to be a key surface through which Scc4 facilitates cohesin loading. Strangely, mutation of five residues within this conserved patch was not lethal (though the documentation here was inadequate). It did however appear to compromise recruitment of Scc2 to centromeres and to reduce the loading of cohesin onto peri-centric DNAs but not onto chromosome arms. Curiously, the anomaly of the most conserved part of Scc4 not being essential for loading cohesin onto chromosomes is not discussed in the manuscript, which is odd. The paper concludes that Scc4's conserved patch is required for peri-centric cohesin loading but not for a second “opportunistic” mode that occurs elsewhere on the chromosome (note that it is this second so called opportunistic mode that seems essential!). The paper also concludes that one of the functions of Scc4 is to mediate Scc2 stabilization, a statement for which there was no adequate justification. Though the structure appears to imply that this would be the case, this should be supported by some biochemical data (see below).

The claim that Scc4's conserved patch is not in fact essential for its function is a surprising one. Bold claims require exceptional documentation and sadly this is lacking here. From the structural figure supplied and the accompanying sequence variation, it would appear that L256, Y298, K299, Y313, F324, K327, and possibly K331 are the key surface conserved residues. And yet, for some reason the mutant Scc4 with the most extensive alterations in its conserved patch was *scc4*^*m35*^, which contained F324A, K324A (presumably this should be K327A -typo?), K331A, K541A, and K542A. In other words, several of the key residues within the patch were left unmutated while residues that appear less conserved (K541 and K542, only conserved in *S. cerevisiae* like yeasts according to the figure) were mutated. If one wished to establish whether or not the conserved patch really is non-essential and has a specialized role in recruiting cohesin to centromeres, then one really needs to obliterate the nature of this surface and see whether the same results are obtained. What if mutating L256, Y298, K299, Y313, F324, K327, and K331 simultaneously were lethal? Would this not greatly alter the thrust of the paper and the interpretation of the results? Would it not imply that in fact, *scc4*^*m35*^ is a hypomorph (with regard to the function of the conserved patch) and that the selective reduction in cohesin loading at centromeres arises because efficient loading at this location is more sensitive to a partial loss of function than loading along chromosome arms? This is a serious issue but not one that would require an inordinate amount of work to address. Making such mutants, characterizing their (non-?) effect on Scc2/4 complex formation in vivo, and a far more careful analysis of their phenotypes rigorously at the genomic level and by live imaging should be a high priority.

Specific comments:

1) The analysis of the effect of mutations within Scc2 that alter conserved Scc4 contact residues (Results) is not state of the art and raises important issues. First of all, the figure is of poor quality. Second, one should not use plasmids to do this type of analysis. As the authors subsequently point out, plasmid borne SCC2 supposedly bypasses the need for SCC4. Why, if the N-terminal domain of Scc2 is concerned solely with Scc4 binding and if plasmid borne SCC2 can suppress lethality due to lack of Scc4, do plasmid born mutant SCC2s that supposedly merely compromise Scc4 binding not suppress lethality due to *SCC2-AID*? Something is badly wrong here, unless there is a misunderstanding of some sort.

Note also that the method of using plasmids is likely to under-estimate the importance of residues that have been mutated. These experiments should be performed by integrating single copy wild type and mutant SCC2 at an ectopic site and testing function by tetrad dissection in a cross heterozygous for an *scc2* deletion. Lastly, it would seem appropriate to test whether the mutations actually affect binding of Scc2's NTD to Scc4. This is easy to do and surprising that this was not done.

2) The claim that SCC2 over-expression bypasses the complete lack of Scc4 is interesting but maybe not too surprising (though see comment 1), but the experiment performed does not prove this. What was shown was that plasmid borne SCC2 can suppress the lethality of GAL-SCC4 cells on glucose. This experiment cannot exclude the possibility that there is minor expression of SCC4 from the GAL promoter and that this contributes to the suppression of lethality. It would be trivial to do this properly, that is, to show that plasmid born SCC2 can suppress the null. This should be done using tetrad analysis not just plasmid shuffle, which can easily lead to the isolation of secondary suppressors.

3) The statement that finding that SCC2 over-expression suppresses the lack of Scc4 is consistent with the essential function of Scc4 is to stabilize Scc2 is fairly meaningless. If the authors think that this is what is going on, then they need to measure protein stability.

4) Results. The authors raise the rather interesting possibility that Scc4's conserved patch might be involved in binding phosphate groups. This is particularly interesting given that the Cdc7 kinase is required for loading cohesin at centromeres. It is surprising that the paper does not address this issue further. Might the conserved patch be involved in binding a phosphorylated target on the Ctf19 complex?

5) Also in the Results. The plasmid mis-segregation assay is not a good one, the data not particularly impressive, and adds very little. These low quality experiments rather detract from the fine structural biology. Why not measure chromosome loss properly if this is so important?

6) In the same section. There appears to be little or no documentation of how the mutations in Scc4's conserved patch were made, how expressed, and how their non-lethality was initially determined. This is crucial part of the paper and the main text must explain carefully how this was done.

7) In the Results. The conclusion that mutations like *scc4*^*m35*^ affect establishment of centromeric cohesion through a Chl4-dependent process is not rigorous. Is it not possible that other hypomorphic mutations of the Scc2/4 complex might have very similar phenotypes according to the experiments described and yet we know that Scc2/4 is required for loading throughout the genome? For example, it would be surprising if scc2-4 would not also have very similar phenotypes on centromeric cohesion. If so, would you conclude that this mutation affected in a specific manner the formation of centromeric cohesion? These results per se do not “strongly suggest that the Scc4 conserved patch targets Scc2/4 specifically to centromeres”. A close look at the data in Figure 4 suggests a slightly longer inter-SBP distance with the double mutant compared to single mutants. But there is nothing to compare a synthetic effect with, such as another mutant that would provide a longer inter-SPB distance when coupled with a *chl4* deletion. The phenotype of *chl4* mutants is stronger than that of *scc4*^*m3*^ mutants and it is therefore hardly surprising that chl4 is epistatic to *scc4*^*m3*^.

Such a result could arise whether or not *scc4*^*m3*^ worked exclusively via Chl4 or not.

8) The experiments claiming that *scc4*^*m35*^ compromises Scc2's recruitment to centromeres and selectively reduces peri-centric cohesin loading are not state of the art. First, there are major problems with analysing the distribution of Scc2 using ChIP. This problem afflicts data from a variety of organisms, not just yeast. The problem is that Scc2 is not stably associated with chromatin like cohesin. The only reliable method of documenting Scc2's localization at centromeres is to use live imaging, which avoids the countless pitfalls associated with ChIP. Fortunately, this can be readily done with *S. cerevisiae* where Scc2-GFP co-localizes with centromeres for much of the cell cycle and the centromeres are conveniently all clustered. The analysis performed here is of very poor quality. The images are not believable. It should be possible, especially if one uses diploids, to observe clear images in all cells post S phase and pre-anaphase. To distinguish mutant and wild type strains, these need to be differentially marked so that they can be placed on the same slide and imaged together and thereby compared rigorously. Moreover, the centromeres need to be marked separately in an unambiguous manner. The second problem concerns the use of QPCR ChIP to analyse cohesin's genomic distribution. In this day and age, it is not sufficient to draw major conclusions about the genomic distribution of proteins without analysing this using ChIP-seq. This would be easy to do.

---

## [Author Response]

*1) More extensive mutagenesis of the surface on Scc4*.

We have constructed strains bearing substitutions at seven of the most conserved amino acid residues that line the surface of the Scc4 conserved patch. We have examined several phenotypes relating to sister chromatid cohesion in these strains. We find that the changes at these seven positions, which extensively modify the wild type protein surface at this site on Scc4, yield strains that are fully viable: the only phenotypes we observe are consistent with a specific inability to enrich cohesin at centromeres and pericentromeres. Moreover, we find that recombinant Scc4-Scc2^1-181^ protein complexes expressed in *E. coli* behave essentially identically regardless of the status of the Scc4 conserved patch. We have included these experiments in the revised version of our manuscript (Figure 1—figure supplement 1; Figure 3—figure supplement 1; Figure 4; Figure 4—figure supplement 2). We conclude that the conserved surface of Scc4 has a specific role in Scc2/4 localization but is not otherwise required for cohesin loading.

*2) Since ChIP with Scc2 is unreliable, results should be backed up by live cell imaging, which is more reliable and a better reflection of what is going on in the cell. This should be done using state of the art imaging (see point 8)*.

To address this point, we have generated new imaging strains and examined Scc2 localization to centromeres in the wild type and Scc4-mutant backgrounds (Figure 4). Consistent with our initial observations, we find that centromeric Scc2-GFP foci are observed only in wild type cells and not in cells with the native copy of *SCC4* replaced by *scc4*^*m7*^. A more detailed discussion of these experiments follows in our response to point 8 of the reviewer's comments, below.

3) Whole genome comparisons of ChIP data would be a more rigorous way of demonstrating that Scc4 is concerned with loading cohesin around centromeres and not along arms. Such a conclusion is hard to make merely on the basis of analyzing a few sequences.

To analyze cohesin localization genome-wide, we have sequenced chromosomal DNA that purifies with Scc1 (ChIP-seq) in strains expressing either wild type or mutant Scc4 from its endogenous locus (Figure 4; Figure 4—figure supplement 3). We find that Scc1 localization across the genome is identical in the wild type and *scc4*^*m35*^ backgrounds with the exception of centromeres and ∼10kb of surrounding DNA on either side. These experiments provide evidence for a specific function of the conserved surface of Scc4 in enrichment of cohesin at centromeric regions.

One of the reviewers had extensive and very helpful comments, to which we respond in detail below.

*Specific comments*:

*1) The analysis of the effect of mutations within Scc2 that alter conserved Scc4 contact residues (Results) is not state of the art and raises important issues. First of all, the figure is of poor quality. Second, one should not use plasmids to do this type of analysis. As the authors subsequently point out, plasmid borne SCC2 supposedly bypasses the need for SCC4. Why, if the N-terminal domain of Scc2 is concerned solely with Scc4 binding and if plasmid borne SCC2 can suppress lethality due to lack of Scc4, do plasmid born mutant SCC2s that supposedly merely compromise Scc4 binding not suppress lethality due to* SCC2-AID*? Something is badly wrong here, unless there is a misunderstanding of some sort*.

There is indeed some misunderstanding. Our experiments show that an increase in *SCC2* copy number suppresses *SCC4* repression and that restoration of normal Scc2 protein rescues *SCC2-AID* depletion. These observations show that Scc4 likely supports the execution of Scc2’s normal function. In the absence of Scc4, more Scc2 is needed to achieve normal function, and with normal levels of Scc2, Scc2-Scc4 contact is required for normal function.

The following comments may further clarify. First, the failure of Scc2^∆138^ to rescue *SCC2-AID* depletion is not total (compare Scc2^∆138^ to vector in Figure 2—figure supplement 2). Second, while transcriptional repression of *SCC4* does not lead to a complete loss of viability, as pointed out by the reviewers, rescue by increasing *SCC2* dosage does not require the Scc2-Scc4 interaction surface, indicating that an increased dosage of complete Scc2/4 complexes is not a likely explanation for this observation.

*Note also that the method of using plasmids is likely to under-estimate the importance of residues that have been mutated. These experiments should be performed by integrating single copy wild type and mutant SCC2 at an ectopic site and testing function by tetrad dissection in a cross heterozygous for an* scc2 *deletion*.

The experiment we present in Figure 2—figure supplement 2 shows that these residues are important, if not essential. We agree with the reviewer that investigation of precise phenotypes arising in cells bearing crippled versions of Scc2 should be the subject of future work, but the data we have presented shows that the interface between Scc2 and Scc4 is critical.

*Lastly, it would seem appropriate to test whether the mutations actually affect binding of Scc2's NTD to Scc4. This is easy to do and surprising that this was not done*.

We attempted to perform Scc2-Scc4 association studies in vitro but were not successful. Closure of the Scc2-Scc4 interaction surface, as the structure shows, probably means that these proteins fold together. The mode of association would in any case complicate measurement of Scc2-Scc4 binding upon mixing of individual recombinant components, because it would be hard to establish a condition of equilibrium.

*2) The claim that SCC2 over-expression bypasses the complete lack of Scc4 is interesting but maybe not too surprising (though see comment 1), but the experiment performed does not prove this. What was shown was that plasmid borne SCC2 can suppress the lethality of GAL-SCC4 cells on glucose. This experiment cannot exclude the possibility that there is minor expression of SCC4 from the GAL promoter and that this contributes to the suppression of lethality. It would be trivial to do this properly, that is, to show that plasmid born SCC2 can suppress the null. This should be done using tetrad analysis not just plasmid shuffle, which can easily lead to the isolation of secondary suppressors*.

The experiment we present in Figure 2—figure supplement 3 shows that compromised Scc4 can be rescued by increased *SCC2* gene dosage and that this rescue does not require the N-terminus of *SCC2*. This result shows that Scc4 plays a supportive role in Scc2 function. While we offer some speculation on this point, the nature of this support should be the subject of future studies.

*3) The statement that finding that SCC2 over-expression suppresses the lack of Scc4 is consistent with the essential function of Scc4 is to stabilize Scc2 is fairly meaningless. If the authors think that this is what is going on, then they need to measure protein stability*.

We have modified the main text so that it reflects our observations more accurately and to make clear that the comment is speculative (please see the subsection headed “Conserved Scc2-Scc4 contacts”).

4) Results. The authors raise the rather interesting possibility that Scc4's conserved patch might be involved in binding phosphate groups. This is particularly interesting given that the Cdc7 kinase is required for loading cohesin at centromeres. It is surprising that the paper does not address this issue further. Might the conserved patch be involved in binding a phosphorylated target on the Ctf19 complex?

We decided that including a thorough discussion of this possibility in our current manuscript would be overly speculative without further evidence for such a mechanism. We have included a brief reference to this model in our updated Discussion section.

*5) Also in the Results. The plasmid mis-segregation assay is not a good one, the data not particularly impressive*, *and adds very little. These low quality experiments rather detract from the fine structural biology. Why not measure chromosome loss properly if this is so important?*

We have moved our previous Figure 2 to Figure 3—figure supplement 1, leaving only one panel for the plasmid loss phenotype in the main figures. This assay, a measure of chromosome loss in a physiological sense, gives a robust measure of the key downstream mitotic outcome of the pathway under investigation.

*6) In the same section. There appears to be little or no documentation of how the mutations in Scc4's conserved patch were made, how expressed, and how their non-lethality was initially determined. This is crucial part of the paper and the main text must explain carefully how this was done*.

We have updated the Methods section of our manuscript to make clearer our strain construction procedures (please see the subsection headed “Yeast strains and culture conditions”).

*7) Results. The conclusion that mutations like* scc4^m35^*affect establishment of centromeric cohesion through a Chl4-dependent process is not rigorous. Is it not possible that other hypomorphic mutations of the Scc2/4 complex might have very similar phenotypes according to the experiments described and yet we know that Scc2/4 is required for loading throughout the genome? For example, it would be surprising if scc2-4 would not also have very similar phenotypes on centromeric cohesion*. *If so, would you conclude that this mutation affected in a specific manner the formation of centromeric cohesion?*

In this manuscript, we provide the first documentation of clear separation of function alleles that are specifically deficient in their inability to designate centromeres as special domains for cohesin loading. We view this as an important contribution, if only because these alleles will allow further investigation of this pathway. Although we suppose it is unlikely that *scc2-4* would show an identical phenotype, this remains to be tested.

*These results per se do not* “*strongly suggest that the Scc4 conserved patch targets Scc2/4 specifically to centromeres*”*. A close look at the data in*
Figure 4
*suggests a slightly longer inter-SBP distance with the double mutant compared to single mutants. But there is nothing to compare a synthetic effect with, such as another mutant that would provide a longer inter-SPB distance when coupled with a* chl4 *deletion. The phenotype of* chl4 *mutants is stronger than that of* scc4^m3^*mutants and it is therefore hardly surprising that chl4 is epistatic to* scc4^m3^*.*

*Such a result could arise whether or not* scc4^m3^
*worked exclusively via Chl4 or not*.

We agree that our *CHL4* epistasis experiments are not conclusive proof that the Scc4 conserved patch and Chl4 function in the same pathway. The revised version of our manuscript reflects this uncertainty (please see the subsection headed “A conserved surface patch on Scc4”). We considered the usefulness of a second “off-pathway” mutant to test the range of signal detectable in in these assays. However, we decided that these experiments would be difficult to interpret given limited knowledge of genetic pathways leading to preferential centromeric cohesin loading and their interplay with DNA replication.

We have repeated our measurements of the spindle pole separation phenotype in triplicate and found no observable difference between the *chl4Δ* and *scc4*^*m3*^
*chl4Δ* strains. An updated figure is included in the current version of the manuscript (Figure 3).

*8) The experiments claiming that* scc4^m35^
*compromises Scc2's recruitment to centromeres and selectively reduces peri-centric cohesin loading are not state of the art. First, there are major problems with analysing the distribution of Scc2 using ChIP. This problem afflicts data from a variety of organisms, not just yeast. The problem is that Scc2 is not stably associated with chromatin like cohesin. The only reliable method of documenting Scc2's localization at centromeres is to use live imaging, which avoids the countless pitfalls associated with ChIP. Fortunately, this can be readily done with* S. cerevisiae *where Scc2-GFP co-localizes with centromeres for much of the cell cycle and the centromeres are conveniently all clustered. The analysis performed here is of very poor quality. The images are not believable. It should be possible, especially if one uses diploids, to observe clear images in all cells post S phase and pre-anaphase. To distinguish mutant and wild type strains, these need to be differentially marked so that they can be placed on the same slide and imaged together and thereby compared rigorously. Moreover, the centromeres need to be marked separately in an unambiguous manner*.

We generated diploid yeast strains carrying homozygous copies of Scc2-GFP and Mtw1-tdTomato for simultaneous imaging of kinetochores and Scc2. Diploid cells showed clear GFP foci in nearly all WT cells we examined. In analogous strains bearing homozygous copies of the *scc4*^*m7*^ allele, we found that these foci were absent or diminished beyond the limit of detection. This observation confirms our hypothesis that Scc4 guides Scc2 localization through its conserved patch.

While we were unable to incorporate a third label to allow imaging of both strains in the same field of view, we performed all imaging experiments by placing both strains immediately next to each other on the same cover slip and imaging during the same session with identical microscope and image collection settings. We also repeated the experiment using independently derived strains for each final diploid (distinct haploid precursors), and we made the same observation each time. We therefore conclude that Scc2 is not efficiently recruited to centromeres in strains bearing the *scc4*^*m7*^ mutation.

*The second problem concerns the use of QPCR ChIP to analyse cohesin's genomic distribution. In this day and age, it is not sufficient to draw major conclusions about the genomic distribution of proteins without analysing this using ChIP-seq. This would be easy to do*.

We carried out ChIP-seq experiments using a 6xHA-tagged version of Scc1 expressed from its endogenous locus. These experiments show that the SCC4 mutations we report specifically impact cohesin recruitment to centromeres and the surrounding regions. We have included these experiments in the revised version of our manuscript (Figure 4; Figure 4—figure supplement 3). We have also included a short note on the implications of these experiments in our updated Discussion.